# Fungal Pathogens Associated with Strawberry Crown Rot Disease in China

**DOI:** 10.3390/jof8111161

**Published:** 2022-11-02

**Authors:** Yanting Zhang, Hong Yu, Meihua Hu, Jianyan Wu, Chuanqing Zhang

**Affiliations:** 1College of Modern Agriculture, Zhejiang Agriculture and Forest University, Hangzhou 311300, China; 2Research Institute for the Agriculture Science of Hangzhou, Hangzhou 310013, China; 3Extension Centre of Agriculture Technology of Zhejiang Province, Hangzhou 310020, China

**Keywords:** strawberry, crown rot disease, fungal pathogens

## Abstract

Strawberry crown rot (SCR) is a serious disease that is generally referred to as seedling anthracnose due to its association with *Colletotrichum* spp. Presently, SCR is the main cause of death of strawberry seedlings. However, management strategies, including fungicides targeting *Colletotrichum* spp., have failed to obtain satisfactory results. Therefore, identifying the exact pathogen species causing SCR could guide its management. A total of 287 isolates were obtained from SCR-diseased plants. Based on the culture, morphology, and phylogenetic characteristics, the above 287 fungal isolates of SCR pathogens were identified as 12 different species, including *Colletotrichum siamense, C. fructicola*, *Fusarium oxysporum*, *F. commune*, *F. equiseti*, *F. solani*, *F. tricinctum*, *Epicoccum sorghinum*, *Stemphylium lycopersici*, *Clonostachys rosea*, *Phoma herbarum,* and *Curvularia trifolii.* Pathogenicity results showed that most isolates were pathogenic to strawberry seedlings and exhibited different degrees of virulence. In severe cases, poor growth on the ground, yellowing of the leaves, and even death of seedlings occurred. In mild cases, only black disease spots appeared on the stems of the strawberry seedlings, and a few withered leaves became necrotic. The inoculation experiments showed that the most virulent species were *C. siamense* and *F. oxysporum*, followed by *F. equiseti*, *P. herbarum*, *Cl. rosea*, *S. lycopersici,* and *C. fructicola,* which had disease incidences above 50%. *E. sorghinum*, *S. lycopersici*, *Cl. rosea*, *P. herbarum* and *Cu. trifolii* were reported to cause SCR for the first time herein. In conclusion, SCR is a sophisticated disease caused by a diversity of pathogenic fungi. This work provides new valuable data about the diversity and pathogenicity of SCR pathogens, which will help in formulating effective strategies to better control of the SCR disease.

## 1. Introduction

Strawberry (*Fragaria* × ananassa Duch.) originates from South America and is the most economically important cultivated small berry plant in the world [1]. Due to its high nutritional value, good economic benefits, short growth cycle, and other characteristics, strawberry is extensively planted on a large scale all over the world. China is currently the world’s largest producer of strawberry, accounting for about one-third of the global output. China is also a major exporter of strawberry and exported 81,100 tons of strawberry in 2017, ranking sixth in the world [2].

In recent years, with the rapid increase in strawberry consumption, the cultivation area has significantly expanded. However, the high temperatures, high humidity, and microclimate environment in greenhouses create favorable conditions for the occurrence of many diseases [3]. Furthermore, after years of continuous cropping, strawberry diseases have obviously increased. Seedling death is the most severe impact that hampers strawberry yields at present in China. It is mainly expressed as crown rot, seedling withering, and, in serious cases, seedling death. These symptoms are consistent with strawberry crown rot (SCR) disease [4]. According to previous reports [5], the cause of death of many strawberry seedlings is ascribed to the root and crown rot of strawberry seedlings. In the early stage of SCR, the disease spot is small with red stripes. It rapidly expands into dark, sunken, and hard disease spots after spreading throughout the plant. When the disease is serious, the pathogen invades the shortened stem and then expands into a ring. The part above the disease spot wilts and dies, and when the humidity is high, fleshy red sticky spores are visible [6]. Some studies and observations found that *Colletotrichum* spp. are associated with these symptoms; therefore, SCR is often referred to as strawberry seedling anthracnose (SSA) [7].

It is now widely believed that strawberry anthracnose (SA), caused by *Colletotrichum* spp., can cause damage throughout the cultivation period of strawberry [8]. When SA occurs, all tissues of strawberry can be infected. It produces typical symptoms on the leaves, stolons, and fruits, such as leaf spot and leaf blight, which are caused by corrosion, fruit rot, branch blight, and other symptoms, lowering the quality of crops [9]. SA also harms strawberry seedlings, creating fouling disease spots on the base of the stem and crown rot [10], which influence plant development. Several species of pathogenic fungi have been reported to cause this disease, including *C. gloeosporioides* complex, *C. acutatum*, complex, and *C. fragariae* [11]. For SA, the *C. acutatum* complex is mainly distributed in South America, North America, and Europe [12]. In China, most pathogenic fungi of strawberry belong to the *C. gloeosporioides* complex [13].

It is very important to identify the pathogenic fungi that cause SCR. Studies in recent years have proved that ITS is highly conserved and cannot accurately identify species of *Colletotrichum* [14]. At present, the main genes used in the molecular identification of *Colletotrichum* include *ACT*, *CHS-I*, *CAL*, ITS, and *GAPDH*. In addition, *C. gloeosporioides* and *C. acutatum*, which were previously identified, have been shown be a composite species [15]. Therefore, combining morphological characteristics with polygenic systematics has become a widely accepted method for the identification of *Colletotrichum* spp. and can resolve the problems existing in the classification to some extent [16]. The translation elongation factor 1-α (*EF-1α*) gene and ITS are commonly used for identification of *Fusarium* sp. [17]. At present, determination of the pathogenic fungi that cause SCR is of particular importance and is a prerequisite to developing the precise prevention and control of SCR, reducing seedling death, and improving the yield of strawberry fruits.

## 2. Materials and Methods

### 2.1. Field Surveys, Disease Symptoms, and Sampling

Field surveys were conducted from 2018 to 2019 throughout important strawberry-growing areas in Zhoushan, Jiande and Yuhang, Zhejiang Province. SCR occurs in the main producing regions and is widely distributed in several varieties [13]. The main strawberry varieties planted were Zhangji, Hongjia, Tianxianzui, Hongyu, and Yuexin. Two greenhouses containing each variety within each collection area were selected, and 10–20 strawberry plants with typical symptoms of SCR were arbitrarily collected in each greenhouse. 

### 2.2. Fungal Isolation

Collected plants showing disease symptoms were washed with running water, and 5 mm stem segments were excised and immersed in 75% alcohol for 30 s, then soaked in 1% sodium hypochlorite solution for 3 min, and washed 3 times with sterile water [11]. The disinfected tissue and conidia solution were poured onto potato dextrose agar (PDA) plates with kanamycin sulfate and streptomycin sulfate (100 mg/L) and cultured at 25 °C for 3 to 5 days [18]. After 2 days, single colonies were selected and cultured on fresh PDA medium. Then, subsamples of all mycelia grown on the PDA were transferred to the new PDA medium for culture for 5 days. After transferring twice, the putative pure colony was transferred to PDA slants, incubated for 5 days, and stored in the refrigerator at 4 °C. All collections were stored at Zhejiang A&F University.

### 2.3. Morphological Characterization

#### 2.3.1. *Colletotrichum* spp.

Morphological and cultural characterizations of *Colletotrichum* spp. were performed according to the methods described by Cano et al. [9]. Each isolate was cultured in triplicate to characterize the aerial hyphae growth and colony culture properties. A Scope A1 optical microscope was used to observe and describe the hypha septa, shape, and color of the spores, and spore germinating mode, among other features. The size of the conidia and appressoria of the pathogenic fungi were measured, and 50 data points of each characteristic were selected for calculation and analysis. The colony color of the aerial hyphae and morphology were recorded as previously described [19,20].

#### 2.3.2. *Fusarium* spp.

Morphological and cultural characterizations of *Fusarium* spp. were performed according to the methods described by Haegi et al. [17]. Colony morphology, macrospore and microspore morphology were observed, and mycelial growth rate and spore size were measured. Each isolate with three replicates and the size of the macrospore and microspore were measured; 50 data points for calculation and analysis were obtained.

#### 2.3.3. Other Fungi

For other fungi, colony morphology was observed; spore size and hyphal growth rate were measured [21]. Each isolate also with three replicates and the data of 50 spores were measured for analysis. By consulting the relevant studies, pathogens were preliminary identified according to their cultivation properties and microscopic characteristics [21].

### 2.4. DNA Extraction, PCR Amplification, and Sequencing

When the mycelia covered two-thirds of the PDA plates (90 mm diameter), the total DNA of the fungal mycelia was extracted with the rapid extraction kit of fungal genomic DNA (Sangon Biological Engineering Co., Ltd., Shanghai, China). The selected target genes of *Colletotrichum* were *ACT*, *CAL*, *CHS-I*, *GAPDH*, and ITS [20,22]. ITS and *EF1-α* were used as target genes to identify *Fusarium* sp. [23], and the target gene selected for other fungi was ITS. Primer sequences selected for PCR amplification are shown in Table 1. Finally, 50 μL of PCR products was obtained by electrophoresis in 1% agarose gel in 1% TAE buffer. EB staining was performed for 30 min, and then photographic mapping was performed under UV light. When the PCR products were detected, the amplicon-free products were retrieved and sequenced directly by Sangon Biological Engineering Co., Ltd. All sequences were grouped by gene type and compared with DNAMAN (V.7.0). Sequences with a single nucleotide polymorphism were amplified and sequenced three times to ensure that the polymorphism was not caused by a sequencing error.

### 2.5. Phylogenetic Analysis

Upstream and downstream primers were selected and manually modified with SeqBuilder software. The referenced standard isolates used in this study are indicated [14]. The downloaded and measured sequences were input into BioEdit V. 7.0.5 in Fasta format, and then the sequence was preliminarily compared with CLUSTAL X software. The preliminary comparison results were manually compared and corrected with BioEdit software. *ACT*, *CAL*, *CHS-1*, *GAPDH*, and ITS were used to identify *Colletotrichum*; ITS and *EF1-α* were used to identify *Fusarium*. Fungi of other genera were identified using ITS. Mega 5.0 software was used to cut the sequence and remove the redundant bases before and after the similar sequences. Finally, the sequence alignment file that was used to build the tree was generated [24]. Gene sequences were compared and corrected using the “W” function in MEGA 5.0. The gene sequences were concatenated. Modeltest 3.7win, win-paup4b10-console, and modeltest2, as implemented in MrMTgui, were used to estimate the best model for nucleotide substitution. Bayesian inference phylogenies were constructed using Mr. Bayes V.3.1.2 [25]. Six simultaneous Markov chains were run for 1,000,000 generations each, and trees were sampled every 100th generation. The first 2000 trees, representing the burn-in phase of the analyses, were discarded, and the remaining 8000 trees were used to calculate posterior probabilities in the majority-rule consensus tree. Phylogenetic trees were drawn using TreeView. The alignments and trees were deposited in TreeBASE.

### 2.6. Pathogenicity Tests

A total of 287 isolates were cultured on PDA with light for 7 days. The surface of the petri dish was submerged in sterile distilled water and scraped with a scalpel to prepare a suspension at a concentration of 2 × 10^6^ conidia/mL. We used a sterilized substrate to plant strawberry seeding in a 10 cm diameter pot; each pot contained only one strawberry seeding; we used 4-week-old strawberry seedlings for pathogenicity experiments.

A 30 mL conidial suspension was evenly applied to the stem base of the strawberry. Sterile water was sprayed as a blank control and each treatment was repeated 10 times. After moisturizing for 48 h, the plants were incubated at 25 °C ± 1 °C and 90% humidity with 14 h light/10 h dark for 3, 7, 10, and 15 days to continuously observe SCR development, including plant incidence, stem discoloration, and spot length [26]. The occurrence of blackening at the base of the stem of the plant or the appearance of a reddish discolored halo or reddish-brown spots in the diseased part indicated that the SCR disease had occurred. The incidence was calculated by dividing the number of infected strawberry plants by the number of inoculated strawberry plants.

According to Koch’s postulates, after the onset of disease, the junction of diseased and healthy tissue was removed, re-isolated, and then purified and cultured in PDA media containing antibiotics. Colony morphology, conidia, and other characteristics were studied.

### 2.7. Statistics and Analysis

For the morphological characteristics of each isolate, we recorded the size and growth rate of the colony and the size of the conidia and appressoria. For the pathogenicity tests, we calculated the disease incidence rate and length of the spots caused by each isolate [26]. One-way analysis of variance (ANOVA) in SPSS Statistics v22 (IBM Corp., Armonk, NY, USA) was performed to assess differences in the extent of vascular discoloration induced by different fungal isolates in all varieties. The homogeneity of variance was assessed using Levene’s test. Tukey’s test was used for comparison of treatment means at the 5% significance level. Two-way ANOVA was performed to determine significant differences among varieties [13,14].

## 3. Results

### 3.1. Field Surveys and Sample Collection

In the greenhouses, strawberry seedling death was extensively observed. For all samples collected, symptoms of black rot were observed at the crown (Figure 1). In the field, the strawberry plants showed wilting and the crown was easily broken (Figure 1a–c). On the crown vascular bundle and stem, brown disease spot appeared (Figure 1d–k).

### 3.2. Fungi Collected

According to the colony morphology of the isolates on the culture medium, the 287 strains isolated from all diseased samples were classified into three main types, namely *Colletotrichum* spp., *Fusarium* spp., and a few other fungi in relatively small numbers (Table 2), with 185 isolates (64.5%), 86 isolates (30.0%), and 15 isolates (5.5%), respectively. 

#### 3.2.1. *Colletotrichum* spp.

The *C. siamense* colonies were villous, with a gray and neat edge in the middle, a yellow edge in the middle of the back, and a dark yellow conidia heap. Conidia a monospora were straight, terete, blunt round at both ends, colorless, and smooth; the appressoria were brown, oval or spindle shaped, with intact margins, and a few were irregularly shaped (Figure 2). *C. fructicola* produced a grayish-black colony with irregular edges, a black to gray margin on its abaxially white back, and a cluster of conidia appearing orange. The conidia were terete and obtuse at both ends or apical at one end; appressoria were single or scattered, mostly light brown to dark brown, and spherical, cylindrical, or spindle shaped, with a few being irregular, and a few produced two subcircular appressoria. Mycelial growth rates, appressoria and conidia sizes are described in detail in Table 3.

For the identification of *Colletotrichum*, the *ACT*, *CAL*, *CHS-1*, *GAPDH*, and ITS regions were successfully amplified and sequenced (Appendix A). Phylogenetic trees were constructed using Bayesian inference (Figure 3). These gene sequences contained 1618 characters after concatenation: *ACT* = 1–202; *CAL* = 203–842; *CHS-1* = 843–1072; *GAPDH* = 1073–1326; and ITS = 1327–1618. All isolates of *Colletotrichum* were identified as the *C. gloeosporioides* complex. Using Bayesian inference, the isolates tested on strawberry were divided into two groups, of which most of the *Colletotrichum* fungi at all sampling sites and on all varieties belonged to *C. siamense*, accounting for 96.93%, and a few strains from Zhangji in Jiande belonged to *C. fructicola*, accounting for 3.07%.

#### 3.2.2. *Fusarium* spp.

All *Fusarium* strains isolated from the strawberry plants had different colony morphologies on PDA. Among them, *F. oxysporum* had aerial hyphae that were woolly, white to pale, looser, colorless, or with a purple matrix, and a hypha diaphragm. The conidia were pedicellate, solitary, pedunculate, and not branched. The large conidia were meniscus- and sickle-shaped, with mostly 3–6 septa. The small conidia were single celled, ovoid to elliptic, and colorless (Figure 4). *F. commune* had abundant airborne mycelia, and was white, dense, and thin. The center of the colony was white and near purple, and the back of the colony was red and purple. Under the microscope, the hyphae were observed to have septa; the spore pedicels were long and branched, and conidia were abundant with colorless single oval or spindle-shaped spores (Figure 4). The *F. equiseti* colonies had lush aerial mycelium that were initially white or pale in color and floccular. There was a gradual change to brown from the center, and the bottom of the initial culture exhibited a yellow to brown gradient. The mycelia were colorless and had a diaphragm. The large conidia were sickle- or meniscus-shaped with 3–5 septa. The small conidia were single-celled, ovoid to elliptic, and colorless (Figure 4). The *F. solani* colony had neat edges, thin and fluffy aerial hyphae, exhibited fast growth, and had white to yellow colonies. The conidia were divided into two forms. The large conidia were spindle-shaped and straight or slightly curved with 3–5 septa, in most cases. The small conidia were elliptic or ovoid with one or no septum. The spores with a thick placenta were spherical, intermediate, or terminal (Figure 4). *F. tricinctum* had a rotate-shaped colony with dense aerated mycelia, a white to red surface, red-yellow mycelia in the center, and were rose-red to dark red on the back of the medium. The conidia were sickle-shaped, relatively straight, and slender, usually with 3–5 septa (Figure 4).

For the identification of *Fusarium*, the ITS and *EF1-α* regions of some *Fusarium* strains were successfully amplified and combined (Appendix A), and phylogenetic trees were constructed by Bayesian inference. The double gene sequence contained 892 characters after concatenation, and the concatenated sequence was ITS = 1–478 and *EF1-α* = 479–892. The strains tested on strawberry were divided into five groups, of which most of the *Fusarium* fungi from Hongjia and Zhangji from Jiande and Zhoushan belonged to *F. oxysporum* with 94% bootstrap support, accounting for 90.48% (Figure 5). The second group of isolates clustered together with the *F. commune* isolates from Hongjia and Zhangji from Zhoushan and Jiande, accounting for 4.76%. A few isolates from Zhoushan and Jiande clustered with *F. equiseti*, *F. solani*, and *F. tricinctum* with 100%, 100%, and 99% bootstrap support, respectively, accounting for 2.38%, 1.19%, and 1.19%, respectively.

#### 3.2.3. Other Fungal Species

In addition to the above two categories of pathogenic fungal species, a few other strains were isolated. The growth rates of these five categories of pathogenic fungi were different on PDA, and the morphological characteristics were also significantly different. According to morphology (Table 3), five species were preliminarily identified: *Epicoccum sorghinum*, *Stemphylium lycopersici*, *Clonostachys rosea*, *Phoma herbarum*, and *Curvularia trifolii* [21].

The *E. sorghinum* colony had dense hyphae that were initially white and later became darker grayish pink or pink with septate mycelia. The conidia were mostly spherical, scattered, or clustered. The thick ascospores were subglobose, square, or elliptic, and dark brown in color. The small conidia were elliptic or oval (Figure 6). *S. lycopersici* had lush and flocculent hyphae. The colonies were pale to pale pink and arranged in a round shape. The back of the petri dish was yellowish-brown with the production of a yellow pigment. The mycelia were thin with partitions, and the conidia were solitary and oval or oval-shaped (Figure 6). The aerial hypha of the *Cl. rosea* colony was relatively lush, showing white to light gray, and the base was brown to black. The hypha had no septum, and the spore pedicels were slender, showing binomial branches. The base of the branches was expanded, and the top produced monoclonal, colorless, elliptic, and large-diameter conidia (Figure 6). The colony morphology of *P. herbarum* was nearly circular, with round lines with uneven edges. The colonies were slightly uplifted at the center, and the hyphae were pale, exhibited vigorous growth, and looked like cotton wool. The surface of the mycelium was rough, not smooth, and exhibited a diaphragm under the microscope, a binary branch, and branch base enlargement. The conidia were ovoid or oblong and colorless (Figure 6). The colonies of *Cu. trifolii* were round, initially grayish white, and then grayish brown to dark bluish brown. The hyphae were dense, and the air-hyphae were underdeveloped, relatively smooth, and abaxially grayish brown. The conidiospore pedicels were solitary, erect, pale brown, and unbranched. The conidia were solitary, with septa (mostly 3–4), medium brown cells, and were near colorless to light brown at both ends (Figure 6).

The ITS dataset included 42 taxa and contained 567 characters with 374 parsimony-informative positions (Appendix A). Analyses of the combined dataset differentiated four well-supported clades (100% bootstrap support), each corresponding to a separate genus: *Epicoccum*, *Stemphylium*, *Clonostachys*, and *Curvularia* (Figure 7A). Within *Clonostachys*, isolates ZS-HX-51-3 and ZS-HX-46-1 from Hongjia in Zhoushan formed a distinct subclade (bootstrap 97%) that clustered next to the *Cl. rosea* subclade (Figure 7A). Within *Epicoccum*, isolates ZS-HX-55-3 and ZS-HX-80 and ZS-HX-84 from Hongjia in Zhoushan formed a distinct subclade (bootstrap 100%) that clustered next to the *E. sorghinum* subclade. Within *Stemphylium*, isolates ZS-HX-72 and ZS-HX-96-2 from Hongjia in Zhoushan formed a distinct subclade (bootstrap 96%) that clustered next to the *S. lycopersici* subclade. Within *Curvularia*, isolate ZS-HX-60-2 formed a distinct subclade (bootstrap 82%) that clustered next to the *Cu. trifolii* subclade (Figure 7A). The *Phoma* combined ITS and EF1-α dataset included 22 taxa and contained 574 characters with 443 parsimony-informative positions. The topology primarily differed in the isolate positions within the *P. herbarum* subclade. All *Phoma* isolates from Zhangji from Zhoushan resided in the *P. herbarum* clade (bootstrap 94%) that clustered next to the *P. macrostoma* subclade (Figure 7B).

### 3.3. Pathogenicity 

In the inoculation experiments, the crown rot caused by 12 different types of pathogenic fungi is shown in Figure 8. Of all the fungi involved in the pathogenicity experiments, *F. asiaticum* was non-pathogenic. The other 12 species were pathogenic. The severity of the crown rot differed between species, and the aggressiveness was significantly different. The aggressiveness tests showed that in Hongjia strawberry, the most virulent species were *C. siamense* and *F. oxysporum*, followed by *F. equiseti*, *P. herbarum*, *S. lycopersici*, and *Cl. rosea*. The incidence of these species reached more than 50%. However, the incidence of *F. commune*, *F. solani*, *F. tricinctum*, and *Cu. trifolii* was less than 50%. (Table 4).

### 3.4. Epidemic Distribution of Pathogenic Fungi

In the epidemic distribution of pathogenic fungi, there were some differences among the different pathogenic fungi in different regions and varieties. Among them, the largest quantity of *C. siamense* was found in all sampled areas, including Zhoushan, Jiande, and Yuhang. It was also found in all sampled strawberry varieties, including Hongjia, Zhangji, Tianxianzui, and Hongyu. Secondly, the prevalence of *F. oxysporum* was also high, and it was isolated in all sampling areas and distributed in all strawberry varieties.

The other strains, which were less abundant, varied in their distribution. *F. commune* was in the Zhoushan and Jiande districts. *E. sorghinum*, *S. lycopersici*, *Cl. rosea*, *P. herbarum*, *F. solani*, *F. tricinctum,* and *Cu. trifolii* were distributed only in the Zhoushan region (Figure 9a). *C. fructicola* and *F. equiseti* were only distributed in the Jiande region. As for the different strawberry varieties, *F. commune* and *F. equiseti* were mainly distributed on both Hongjia and Zhangji. *E. sorghinum*, *S. lycopersici*, and *Cu. trifolii* were distributed only in the Hongjia variety, and *Cl. rosea*, *P. herbarum*, *F. solani*, and *F. tricinctum* were found only in Zhangji (Figure 9b).

## 4. Discussion

With the rapid increase in strawberry consumption in recent years, the development of the strawberry cultivation industry has accelerated, and the planting regions of strawberry have increased [27]. However, the disease is becoming more and more serious due to continuous cropping [4]. Among the diseases impacting strawberry, SCR, which often occurs in the seedling stage, constitutes a major obstacle that has been difficult to solve. Studies have shown that SCR diseases mostly occur at the root and stem base [28]. The disease becomes serious when the aerial parts of strawberry leaves turn yellow, wither, and die [29], causing seedling death. Studies have shown that not only *Colletotrichum* spp. can cause SCR [5]. There have been reports that strawberry plants can be infected with one or more pathogenic fungi, including *F. oxysporum*, *Macrophomina phaseolina*, *Phymatotrichopsis omnivora* [30].

In China, SCR often occurs in the seedling stage, causing damage to the stem. This is frequently associated with *Colletotrichum* spp., and therefore SCR is often mistaken for anthracnose [7]. Studies have found that crown rot, and not just seedling anthracnose, are responsible for serious strawberry deaths at the seedling stage in recent years, with differences in symptoms and pathogens [31]. At present, in the field, SCR control is only aimed at *Colletotrichum* spp., with more priority given to chemical control [32]. As a result of other pathogens causing SCR as well as the differences between different regions, the efficacy of chemical control in the field is not ideal [33]. It has also been shown that strawberry seedling death cannot be comprehensively understood or prevented if viewed as seedling anthracnose. When comprehensive prevention and control measures are being developed and the agents to be used in the field are being determined, the main pathogenic fungi of SCR should be identified first so that the appropriate chemical can be used to control the disease and promote efficient prevention and control.

In this study, 287 fungal isolates were obtained from strawberries showing SCR symptoms in Zhoushan, Jiande, Yuhang, and other major strawberry production areas were identified as 12 different species. Among them, *C. siamense*, *F. oxysporum*, *E. sorghinum,* and *C. fructicola* accounted for a higher proportion of all strains, comprising 63.07%, 26.48%, 2.09%, and 1.39%, respectively. The pathogenicity experiments showed that *C. siamense*, *C. fructicola*, *F. oxysporum*, *F. equiseti*, *F. solani*, *E. sorghinum*, and *Cu. trifolii* caused symptoms of SCR and blackening. The incidence of *C. siamense*, *F. oxysporum*, *F. equiseti*, *P. herbarum*, *S. lycopersici*, and *Cl. rosea* was relatively high; the disease incidence reached more than 50%, which indicated that these species were the main fungi causing SCR. This study confirmed the diversity of pathogenic fungi for SCR, which included not only *Colletotrichum* and *Fusarium*, as previously reported, but also *P. herbarum*, *S. lycopersici*, and *Cl. rosea*, which were first reported on strawberry stems. This study shows that the number and variety of pathogenic fungi causing SCR were more diverse, and the symptoms of the disease were more complex and varied.

For *Colletotrichum*, the phenomenon of root and crown rot occurred after the infection of strawberry, and with the duration of infection, disease also occurred on the leaves, petioles, flowers, and fruits, as has been reported in China and in other countries [7]. In 2019, *C. siamense* was first reported on strawberry plants in Zhejiang Province and caused disease of the leaves and fruits [34]. Hang et al. (2014) confirmed that *C. siamense* caused strawberry root and SCR through multi-gene identification [35]. *C. fructicola* in Hubei was more pathogenic to strawberry than *C. gloeosporioides* and *C. aenigma*. A study on the population diversity of strawberry *Colletotrichum* reported that *C. fructicola* was the dominant species [34], while *C. siamense* was the second dominant species in this study. At present, the main genes used in the molecular identification of *Colletotrichum* include *ACT*, *CHS-I*, *CAL*, *TUB2*, and *GAPDH*. Studies in recent years have shown that the ITS region is highly conserved and cannot accurately identify the species of *Colletotrichum* [15]. In addition, *C. gloeosporioides* and *C. acutatum* were previously identified to be a species complex [36]. For example, *C. gloeosporioides* represents both a single species as well as a species complex consisting of 22 species plus a subspecies, which are closely related to each other, morphologically diverse, have conidia with similar morphology and ITS sequence differences, and have a wide range of heritability and biological diversity. As a result, *C. aenigma*, *C. conoides*, *C. fructicola*, *C. gloeosporioides*, *C. siamense*, *C. wuxiense*, and *C. tropicale*, among others, were documented and reported [22,23]. Therefore, the identification of *Colletotrichum* by combining morphological characteristics with polygenic systematics has become a widely accepted method for the identification of *Colletotrichum* spp. and could solve classification problems to some extent [15,34]. Using five genes, *Colletotrichum* isolates from strawberry in Hubei province were identified as *C. gloeosporioides*, *C. aenigma*, *C. murrayae*, and *C. fructicola*, and simultaneously the three strains were identified as *C. nymphaeae* in the *C. acutatum* complex. In this study, based on *ACT*, *CAL*, *CHS-1*, *GAPDH*, and ITS, crown rot sample separation was used for genetic identification to improve the identification accuracy of *Colletotrichum*. The results showed that most *Colletotrichum* belonged to *C. siamense*, accounting for 96.93% of the total number of *Colletotrichum* spp. A few strains belonged to *C. fructicola*, accounting for 3.07%. The pathogenicity studies also showed that *C. siamense* was more aggressive than *C. fructicola* and caused more severe damage to the plant stem. This differs from the results of most previous studies, but it is still reasonable to speculate that *C. siamense* may be the dominant species of *Colletotrichum* causing SCR. Thus, subsequent experiments are needed to prove this conclusion.

Strawberry diseases caused by *Fusarium* are common across the world. *F. oxysporum* has been widely reported as the main pathogenic fungus of strawberry root rot in recent years [37]. In 2016, it was reported in Iran that *Fusarium,* including *F. oxysporum*, *F. solani*, *F. acuminatum,* and *F. equiseti*, infected strawberries, causing SCR and fruit rot, among which *F. oxysporum* f. sp. *fragariae* accounted for most isolates and were the most aggressive [38]. In 2014, *F. solani* was also discovered for the first time to cause SCR and blackening or stunting in Spain [39]. This is consistent with the phenomenon of deadly SA studied herein. The major gene loci currently used in the identification of *Fusarium* are rDNA-ITS, *EF1-α*, *mtSSU*, and *β-tubulin* [38]. The *EF1-α* gene, as a DNA barcode, can ensure a high level of amplification and sequencing success and is better able to distinguish between *Fusarium* spp. [40]. Therefore, in the present study, the ITS combined with the *EF1-α* gene were selected to conduct polygenic identification of *Fusarium*, and the results showed that *F. oxysporum* accounted for the majority, followed by *F. commune*, *F. equiseti*, *F. solani*, and *F. tricinctum*. The aggressiveness of *F. oxysporum* was comparatively higher, and the incidence on strawberry seedlings was relatively high, followed by *F. equiseti*, *F. solani*, and *F. tricinctum*, whereas *F. asiaticum* was not pathogenic. The results showed that *F. oxysporum* was the main pathogen of SCR. This is consistent with previous studies [38,39]. Moreover, this is the first time that *F. commune* and *F. tricinctum* were found to cause root crown decay in strawberry plants, and the virulence test showed that *F. equiseti* was second, following *F. oxysporum*.

In addition to the above two main fungal genera that have been widely reported to infect strawberry and cause crown rot, there have been reports about other fungal genera causing crown rot of strawberry in recent years. For example, in strawberry production areas, such as Spain and North America, reports showed that *Phytophthora* spp., including *P. cactorum* and *P. nicotianae*, can cause strawberry root and crown rot and stem break [41]. In 2013, *Pythium helicoides* was also isolated and identified as a pathogen of strawberry root rot and crown rot in several strawberry growing areas in Japan, where it is widely distributed [42]. Another study found that *Macrophomina phaseolina* caused crown rot on strawberries in Chile [29]. Zhang et al. (2018) first discovered the phenomenon of SCR and plant wilt caused by *Nectria pseudotrichia* in China [43]. Several other fungal species have been found during the isolation of pathogens of SCR. After ITS sequence comparison, these strains were identified as *E. sorghinum*, *S. lycopersici*, *Cl. rosea*, *P. herbarum*, and *Cu. trifolii*. Although the isolation rate of these strains was not high, some strains, such as *P. herbarum*, *Cl. rosea*, and *S. lycopersici*, were highly pathogenic, with a disease incidence of up to 60%. Among them, *Cl. rosea*—also a biological agent—had good prevention and control concerning strawberry gray mold under field conditions and greatly reduced the incidence of gray mold on the leaves, fruits, and flowers, and improved the yield of strawberry [44]. This is the first time that these fungi have been reported to cause rot and blackening of strawberry stems. Some fungi, such as *E. sorghinum*, *S. lycopersici*, and *Cu. trifolii*, were reported as strawberry plant pathogens for the first time. However, the biological characteristics and pathogenic mechanisms should be further studied.

Regarding the distribution of pathogenic fungi, there were differences among different types of pathogenic fungi in different regions and varieties. *C. siamense* and *F. oxysporum* were the most abundant, which were distributed in all sampled areas, including Zhoushan, Jiande, and Yuhang. They were also found in all sampled strawberry varieties, including Hongjia, Zhangji, Tianxianzui, and Hongyu, indicating that these two species were widely distributed in regions and varieties. This is consistent with several previous research conclusions [34,38]; although the microbial community composition is affected by the region, strawberry variety, strawberry growth period, and other factors, it can be speculated that *C. siamense* and *F. oxysporum* are the main pathogens of SCR in Zhejiang Province. The number of other species was relatively small and the distribution was diverse. Among the different strawberry varieties, the pathogenic fungi on Zhangji and Hongjia were more abundant, while those on Tianxianzui, Hongyu, and Yuexin were less abundant. Among them, only *C. siamense* was isolated from Yuexin strawberry. The cultivation area in Zhejiang Province is increasing each year and the cultivation duration is longer, but due to heat, the ability of strawberries to adapt to humidity is poor during the summer. Seedlings are affected by high temperatures and rainy climate conditions, which lead to death and seriously impact strawberry production [11,34]. As new varieties, Hongyu and Yuexin are often used as supplementary varieties for their good growth and strong disease resistance [45]. Therefore, it can be reasonably speculated that Zhangji and Hongjia are more susceptible to pathogen infection of crown rot, while other varieties show more resistance and can be expanded in strawberry production. There were more species of pathogenic fungi in Zhoushan and Jiande, while there were fewer in Yuhang, which contained only *C. siamense* and *F. oxysporum*. The reasons for this distribution may be due to the different geographical locations and climatic conditions of the sampling sites. Zhoushan is located in the Yangtze River Delta region. The soil is rich in trace elements and is thus suitable for many pathogenic fungi. Jiande is the most important cultivation area of strawberry in Zhejiang Province, with more than 20 varieties planted [46].

The above studies showed that the diversity of SCR pathogenic fungi in different regions was not limited to *Colletotrichum*. With the changes in regional climate and environment, the pathogenic species also showed obvious changes. Our results demonstrate that it is not sufficient to target *Colletotrichum* as the sole pathogen or to control SA only at the seedling stage. To summarize, this study identified pathogens causing SCR based on the combined analysis of morphology, molecular biology, and pathogenicity tests. SCR pathogenic fungi species include not only *Colletotrichum* and *Fusarium* spp. but also other fungi, such as *E. sorghinum*, *S. lycopersici*, and *Cu. trifolii*. Furthermore, this study discussed the different types of crown rot fungi in terms of the region, variety, and disease symptoms in the field, thereby elucidating the various types of SCR fungi and providing guidance and help for the early diagnosis, prediction, and prevention of SCR.

## 5. Conclusions

It is not correct to regard even managed strawberry crown rot (SCR) as seedling anthracnose due to its association with *Colletotrichum* spp. SCR is caused by compound infection of different species of pathogenic fungi. Among them, *E. sorghinum*, *P. herbarum*, *Cu. trifolii*, *S. lycopersici*, and *Cl. rosea* were reported to cause SCR for the first time.

## Figures and Tables

**Figure 1 jof-08-01161-f001:**
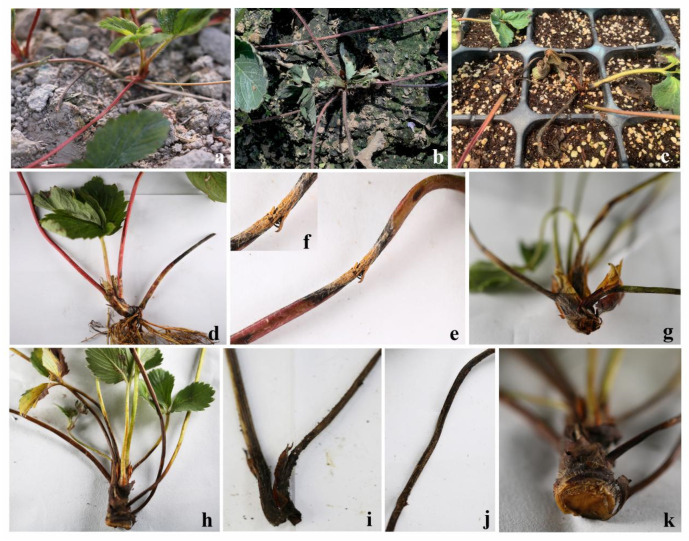
Symptoms and disease characteristics of strawberry crown rot. (**a**–**c**), crown rot of strawberry in the field. (**d**–**k**), crown show dry and brown.

**Figure 2 jof-08-01161-f002:**
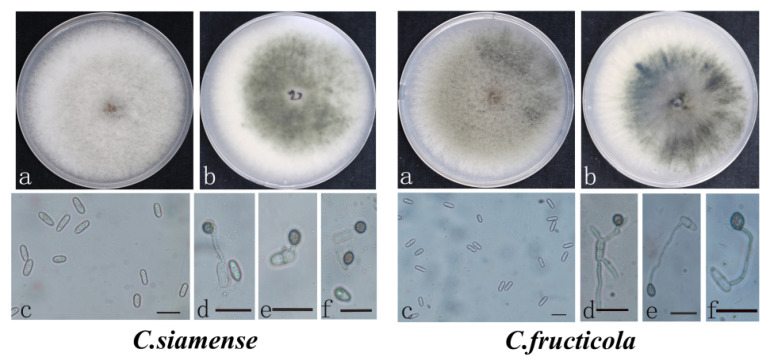
The morphological characteristics of *Colletotrichum siamense* and *C. fructicola* causing strawberry seedling crown rot. *C. siamense* (**left**) and *C. fructicola* (**right**) on PDA (**a**,**b**), conidia (**c**), and appressoria (**d**–**f**). Scale = 20 µm.

**Figure 3 jof-08-01161-f003:**
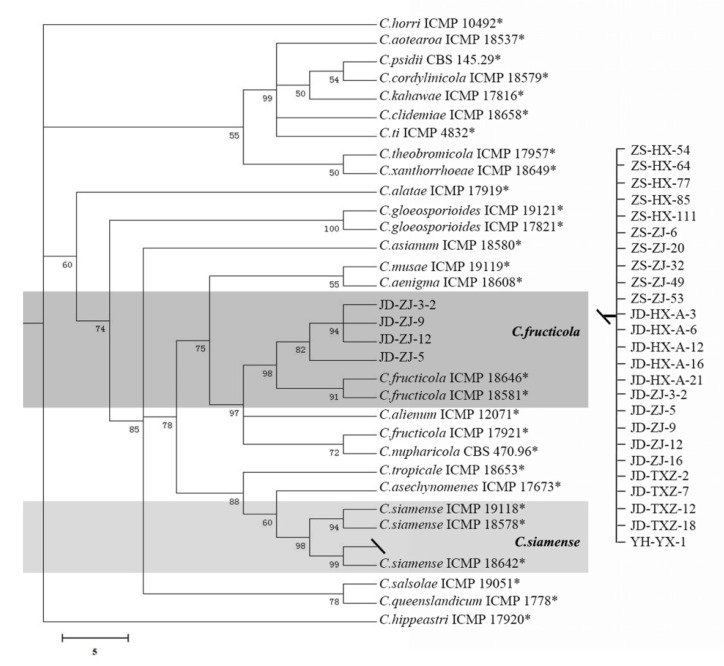
Bayesian inference phylogenetic tree of the *Colletotrichum* spp. isolated from strawberry.

**Figure 4 jof-08-01161-f004:**
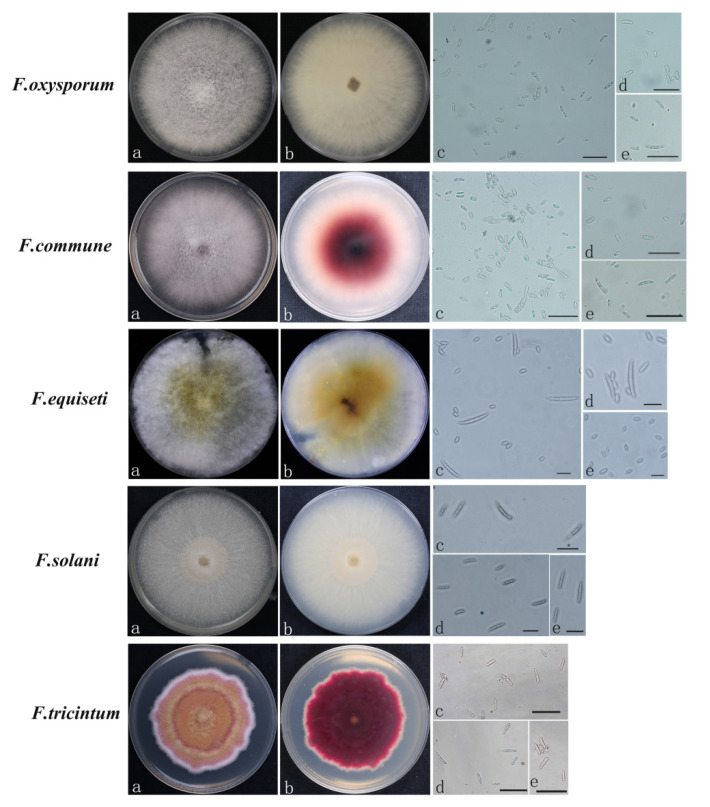
The morphological characteristics of *Fusarium* causing strawberry seedling crown rot. *F. oxysporum*, *F. commune*, *F. equiseti*, *F. solani*, and *F. tricinctum* on PDA (**a**,**b**), conidia (**c**–**e**). Scale = 20 µm.

**Figure 5 jof-08-01161-f005:**
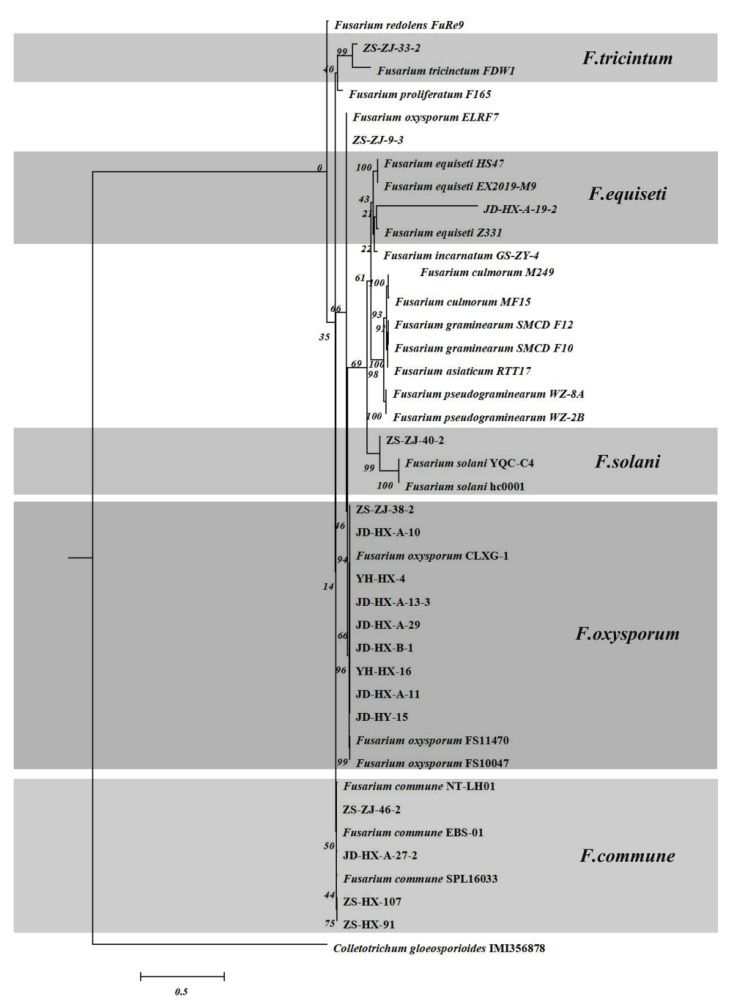
Bayesian inference phylogenetic tree of the *Fusarium* spp. isolated from strawberry.

**Figure 6 jof-08-01161-f006:**
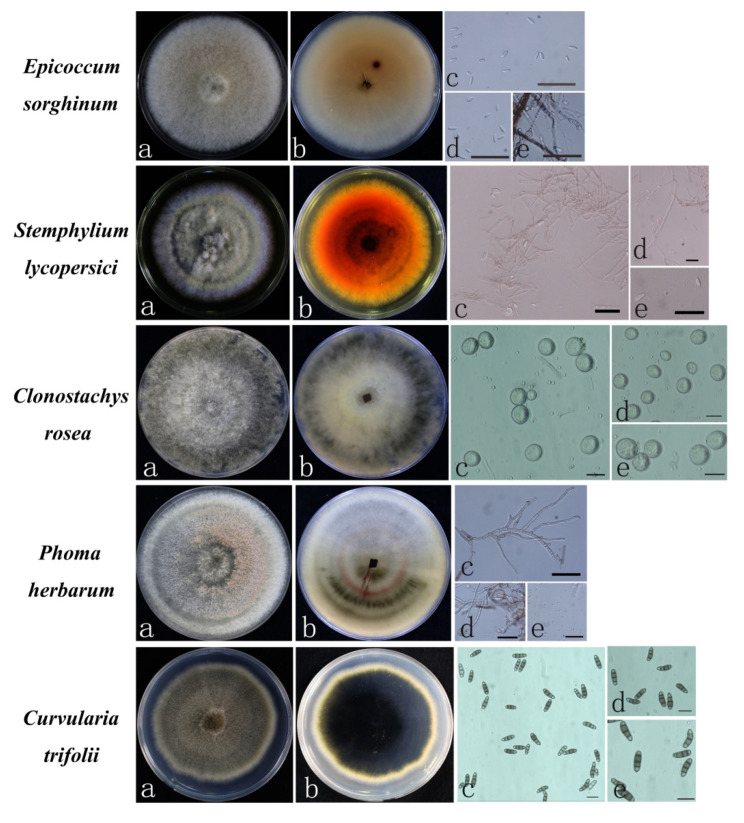
The morphological characteristics of other fungi causing strawberry seedling crown rot. *E. sorghinum*, *S. lycopersici*, *Cu. trifolii*, *P. herbarum*, and *Cl. rosea* on PDA (**a**,**b**), conidia (**c**–**e**). Scale = 20 µm. Bayesian inference phylogenetic tree of other fungi isolated from strawberry.

**Figure 7 jof-08-01161-f007:**
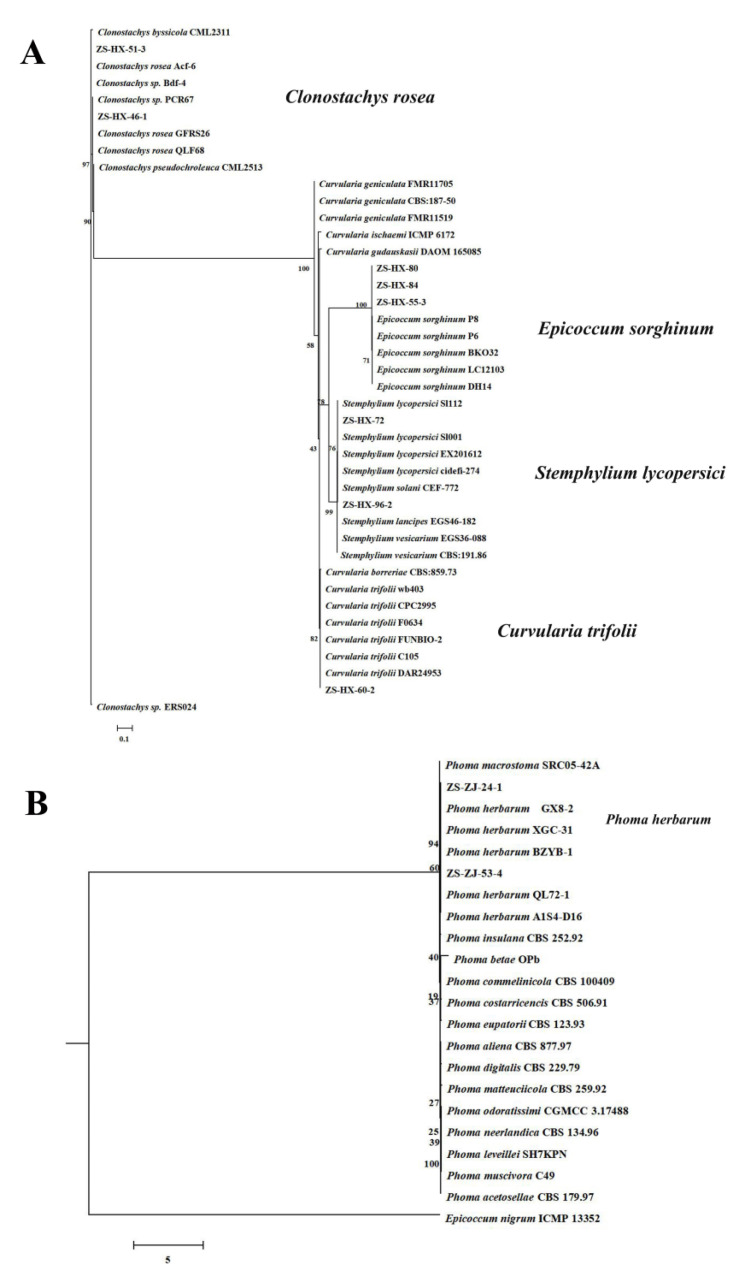
Bayesian inference phylogenetic tree of other fungi isolated from strawberry. (**A**) other species; (**B**) *Phoma herbarum*.

**Figure 8 jof-08-01161-f008:**
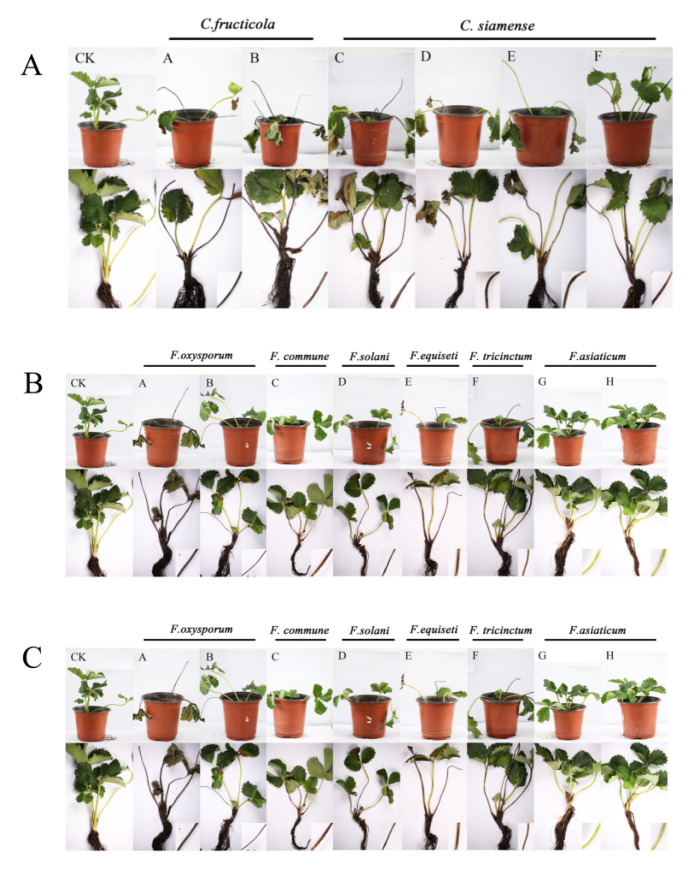
The pathogenicity of *Colletotrichum* spp. (**A**), *Fusarium* spp. (**B**), and other fungi (**C**).

**Figure 9 jof-08-01161-f009:**
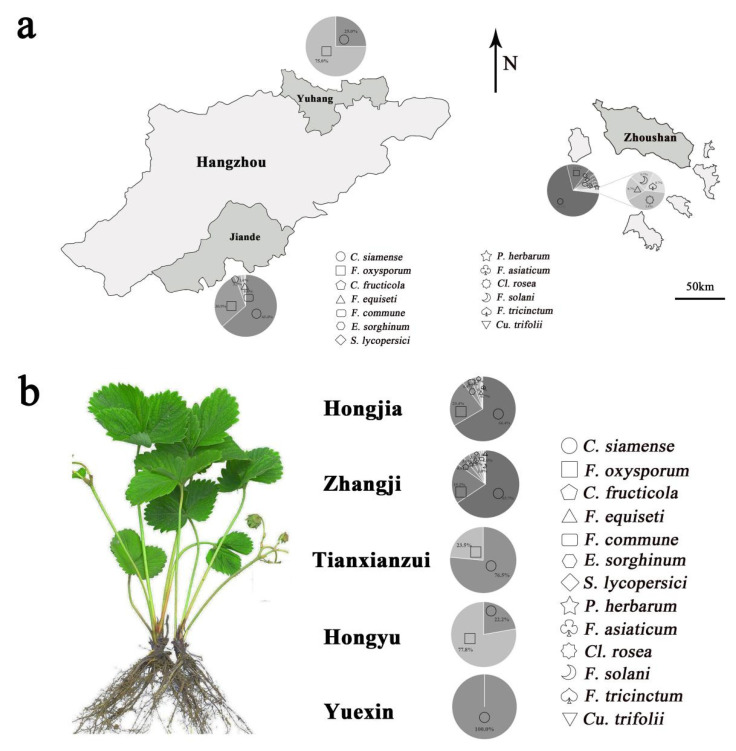
Distribution differences of crown rot pathogens in different regions (**a**) and strawberry varieties (**b**).

**Table 1 jof-08-01161-t001:** Primers used in this study.

Species	Gene	Primers	Direction	Length (bp)	Sequence
*Colletotrichum* spp.	ACT	ACT-512F	Forward	399	ATGTGCAAGGCCGGTTTCGC
		ACT-783R	Reverse		TACGAGTCCTTCTGGCCCAT
	CAL	CL1C	Forward	773	GAATTCAAGGAGGCCTTCTC
		CL2C	Reverse		CTTCTGCATCATGAGCTGGAC
	CHS-I	CHS-79F	Forward	779	TGGGGCAAGGATGCTTGGAAGAAG
		CHS-354R	Reverse		TGGAAGAACCATCTGTGAGAGTTG
	GAPDH	GDF	Forward	870	GCCGTCAACGACCCCTTCATTGA
		FDR	Reverse		GGGTGGAGTCGTACTTGAGCATGT
*Fusarium*	EF1-α	EF1	Forward	680	ATGGGTATAAGGAAGGACAAGAC
		EF2	Reverse		GGAGAGTACCAGTGAATCATGTT
other species	ITS	Its-1F	Forward	593	CTTGGTCATTTAGAGGAAGTAA
		Its-4R	Reverse		TCCTCCGCTTATTGATATGC

**Table 2 jof-08-01161-t002:** The number of fungal isolates obtained from strawberry plants showing disease symptoms.

Fungal Species	Strawberry Varieties					
	Hongjia	Zhangji	Tianxianzui	Hongyu	Yuexin	Total
*Colletotrichum* spp.	91	69	13	6	6	185
*Fusarium* spp.	36	25	4	21	6	86
Other species	10	5	0	0	0	15
Total	137	99	17	27	6	287

**Table 3 jof-08-01161-t003:** Conidia and appressoria morphology and colony growth rate for 12 pathogenic fungal species.

Species ^v^	Conidia		Appressoria		Growth Rate (mm/ day) ^x^
	Length (μm)	Width (μm)	Length (μm)	Width (μm)	
*C. fructicola*	17.18 ± 0.23	6.21 ± 0.19	8.19 ± 0.10	6.50 ± 0.13	14.18 ± 0.02
*C. siamense*	15.07 ± 0.49	6.26 ± 0.26	7.09 ± 0.11	6.46 ± 0.09	13.51 ± 0.17
*F. oxysporum*	15.49 ± 0.71	5.96 ± 0.27	11.35 ± 0.20	5.73 ± 0.15	13.72 ± 0.31
*F. commune*	13.71 ± 0.31	5.12 ± 0.09	8.75 ± 0.20	4.11 ± 0.08	13.91 ± 0.30
*F. equiseti*	37.90 ± 3.04	7.30 ± 0.55	11.93 ± 0.32	7.26 ± 0.20	14.52 ± 0.27
*F. solani*	13.53 ± 0.26	6.48 ± 0.16	7.99 ± 0.13	4.85 ± 0.08	14.48 ± 0.19
*F. tricinctum*	15.92 ± 0.71	5.06 ± 0.13	/	/	9.43 ± 0.07
*E. sorghinum*	16.39 ± 0.32	7.91 ± 0.18	/	/	13.85 ± 0.11
*S. lycopersici*	6.43 ± 0.22	3.60 ± 0.09	/	/	12.83 ± 0.08
*Cl. rosea*	22.74 ± 0.37	21.13 ± 0.28	/	/	13.77 ± 0.03
*P. herbarum*	19.81 ± 0.95	7.74 ± 0.22	/	/	12.70 ± 0.20
*Cu. trifolii*	25.05 ± 0.38	9.42 ± 0.17	/	/	12.36 ± 0.18

^v^*Colletotrichum siamense*, *C. fructicola*, *Fusarium oxysporum*, *F. commune*, *F. equiseti*, *F. solani*, *F. tricinctum*, *E. sorghinum* (*Epicoccum sorghinum*), *S. lycopersici* (*Stemphylium lycopersici*), *Cl. rosea* (*Clonostachys rosea*), *P. herbarum* (*Phoma herbarum*), *Cu. trifolii* (*Curvularia trifolii*). **^x^** Data are the mean ± standard error.

**Table 4 jof-08-01161-t004:** Disease incidence, and lesion length for 12 pathogenic fungal species.

Species ^v^	Pathogenicity	
	Disease Incidence (%)	Lesion Length (mm)
*C. fructicola*	39.34 ± 1.40 d	92.13 ± 3.98 a ^z^
*C. siamense*	63.49 ± 1.58 a	64.00 ± 3.79 bc
*F. oxysporum*	63.17 ± 2.37 a	75.83 ± 6.76 ab
*F. commune*	38.96 ± 3.16 d	69.77 ± 8.51 b
*F. equiseti*	52.38 ± 2.75 bc	40.40 ± 3.83 de
*F. solani*	39.54 ± 2.88 d	40.00 ± 4.83 de
*F. tricinctum*	28.33 ± 2.44 e	21.33 ± 3.84 ef
*E. sorghinum*	45.93 ± 6.18 cd	42.22 ± 5.63 de
*S. lycopersici*	50.49 ± 3.75 bc	59.00 ± 2.39 bc
*Cl. rosea*	52.91 ± 3.26 bc	42.67 ± 3.93 de
*P. herbarum*	57.14 ± 2.13 ab	66.09 ± 5.93 b
*Cu. trifolii*	16.09 ± 1.74 f	13.40 ± 1.03 f

^v^ The strains for pathogenicity assays in this study. ^z^ Data are mean ± standard error. Mean values with the same letters were not statistically different (*p* > 0.05) according to the least significant difference (LSD) test.

## Data Availability

Sequences have been deposited in GenBank (Appendix A). The data presented in this study are openly available in NCBI. Publicly available datasets were analyzed in this study. These data can be found here: https://www.ncbi.nlm.nih.gov/.

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
