# Peer review of "Fungal Pathogens Associated with Strawberry Crown Rot Disease in China"

_jof, 2022, doi:10.3390/jof8111161_

Round 1

Reviewer 1 Report

Dear authors, although I appreciate the amoun of work done, with regret, I have to say that the manuscript presents some inconsistencies. The introduction is focused on Colletotrichum while the disease seems related to a group of different pathogens. The Materials & Methods are poorly described (e.g. the inoculation method) and overall the scientific style and English must be improved. I am not satisfied by the fungal identification part as the genes/regions selected for Colletotrichum identification seem not sufficient (a beta-tub2 or ApMat would be probably needed) and ITS is surely not sufficient for other genera (e.g. Stemphylium). This is particularly relevant if you want to conduct also phylogenetic studies. As such, in my opinion, the manuscript requires further experiments not compatible with the time of a review.

Author Response

Dear authors, although I appreciate the amoun of work done, with regret, I have to say that the manuscript presents some inconsistencies. The introduction is focused on Colletotrichum while the disease seems related to a group of different pathogens.

Response: The CR disease was called as seedling anthracnose and mange this disease as anthracnose caused by Colletotrichum. This study indicated that the CR disease related to a group of different pathogens.

 The Materials & Methods are poorly described (e.g. the inoculation method) and overall the scientific style and English must be improved.

Response: This was re-wrote.

I am not satisfied by the fungal identification part as the genes/regions selected for Colletotrichum identification seem not s.ufficient (a beta-tub2 or ApMat would be probably needed) and ITS is surely not sufficient for other genera (e.g. Stemphylium). This is particularly relevant if you want to conduct also phylogenetic studies. As such, in my opinion, the manuscript requires further experiments not compatible with the time of a review.

Response: Dear reviewers, I have modified the manuscript according to your requirements and suggestions. For Colletotrichum , especially for C. gloeosporioides complex and only Colletotrichum siamense, C. fructicola in this study. at present, the main locus used  in the molecular identification of Colletotrichum were ACT, CHS-I, CAL, TUB2, ITS, and GAPDH.

Reviewer 2 Report

This work provides new valuable data about the diversity and pathogenicity of SCR pathogens, which will help in formulating effective strategies to better control of the SCR disease. However, all the manuscripts need some improvement. Many, in the quality of dendrograms, and scientific language. The authors should use the same word/term for the same thing..e.g., variety or cultivar! I've highlighted in the PDF.

The title of the manuscript could be improved..

Author Response

This work provides new valuable data about the diversity and pathogenicity of SCR pathogens, which will help in formulating effective strategies to better control of the SCR disease. However, all the manuscripts need some improvement. Many, in the quality of dendrograms, and scientific language. The authors should use the same word/term for the same thing..e.g., variety or cultivar! I've highlighted in the PDF.

The title of the manuscript could be improved..

Response: Thanks for your recognition of the content of the article. I have modified the text according to the requirements and suggestions in the PDF, including figure, grammar et al. And the title of the manuscript also has been revised. Hopefully, this study is updated and strong enough to be accepted by the Journal of Fungi.

The title was changed as: Fungal pathogens associated with strawberry crown rot disease in China.

Reviewer 3 Report

good job

Author Response

Thanks for your suggestions and courage, we improved our ms. according comments of all the reviewers and editors.

Reviewer 4 Report

Dear authors,

The manuscript jof-1940083, entitled "Strawberry crown rot disease caused by compound infection of twelve different fungal species", submitted to Journal of Fungi has been reviewed. I noticed a number of grammatical errors throughout the text but there are many more; at some point I just stopped listing them. Therefore, I invite you to send the new manuscript to a native speaker to prepare a better version. In general, results are based on sound experiments with appropriate methodologies. However, in order to improve the manuscript, some suggestions along with the needed revisions have been made in the text of the article (attached file), which should be considered by the respected authors. Regarding to this manuscript the following points should be considered by the authors. In addition, for all my comments and more details, please check the attached pdf file. 

Some general issues:

1-Title:

The title of the manuscript should be changed according to the content of the manuscript. This title may be better for the manuscript: Fungal pathogens associated with strawberry crown rot disease in China

 2-Abstract:

This part seems a bit confusing. In this section, it is better to give the results of the species identification, and then the results of the pathogenicity tests.

 3-Materials and Methods:

In general, in this section, the isolation of fungi and their morphological and molecular identification should be written at first, and after the identification, the pathogenicity test should be given.

 4-Results:

The order of the headings written in the results section should be consistent with the Materials and Methods (The order of headings and their descriptions in the results section should be based on the order specified for Materials and Methods).

Microscopic figures are not of good quality

All figures must have their own captions (as shown in the pdf file)

Phylogeny trees are not captioned. Phylogeny results should also be well interpreted in the text.

For pathogenicity of the identified fungal species, the amount of f and P value should be determined.

The title of the tables need to be revised.

 5-References:

Some references need minor revisions (based on the attached PDF file).

I invite the authors to provide all these corrections in the entire body of the manuscript to make this study updated and strong enough to be accepted in Journal of Fungi.

Good luck in your new study,

Author Response

Dear authors,

The manuscript jof-1940083, entitled "Strawberry crown rot disease caused by compound infection of twelve different fungal species", submitted to Journal of Fungi has been reviewed. I noticed a number of grammatical errors throughout the text but there are many more; at some point I just stopped listing them. Therefore, I invite you to send the new manuscript to a native speaker to prepare a better version. In general, results are based on sound experiments with appropriate methodologies. However, in order to improve the manuscript, some suggestions along with the needed revisions have been made in the text of the article (attached file), which should be considered by the respected authors. Regarding to this manuscript the following points should be considered by the authors. In addition, for all my comments and more details, please check the attached pdf file. 

Some general issues:

1-Title:

The title of the manuscript should be changed according to the content of the manuscript. This title may be better for the manuscript: Fungal pathogens associated with strawberry crown rot disease in China

Response: Thank you for your advice, the title has been revised.

 2-Abstract:

This part seems a bit confusing. In this section, it is better to give the results of the species identification, and then the results of the pathogenicity tests.

Response: OK, this part has been adjusted. the morphological and molecular identification at first, then pathogenicity test.

 3-Materials and Methods:

In general, in this section, the isolation of fungi and their morphological and molecular identification should be written at first, and after the identification, the pathogenicity test should be given.

Response: OK, this part has been adjusted. the isolation of fungi at first, then morphological and molecular identification, finally pathogenicity test.

 4-Results:

The order of the headings written in the results section should be consistent with the Materials and Methods (The order of headings and their descriptions in the results section should be based on the order specified for Materials and Methods).

Microscopic figures are not of good quality

All figures must have their own captions (as shown in the pdf file)

Phylogeny trees are not captioned. Phylogeny results should also be well interpreted in the text.

For pathogenicity of the identified fungal species, the amount of f and P value should be determined.

The title of the tables need to be revised.

Response: I have modified Figure 2-4 ,title et al. according to your suggestion and PDF. The order of the headings written in the results section is adjusted according to the material and method.

 5-References:

Some references need minor revisions (based on the attached PDF file).

Response: I have modified the format of the references according to the PDF.

I invite the authors to provide all these corrections in the entire body of the manuscript to make this study updated and strong enough to be accepted in Journal of Fungi.

Good luck in your new study,

Response: I have modified the manuscript according to the requirements and suggestions in the PDF file. Hopefully, this study is updated and strong enough to be accepted by the Journal of Fungi.

Round 2

Reviewer 1 Report

Although some modifications to the manuscript were carried out, the main methodological issues that I raised were not addressed by authors as such my recommendation cannot change

Author Response

The research design, methods and results were adjusted as suggested.

Reviewer 4 Report

Dear Authors,

In the revised version of the manuscript, many suggestions and comments have been applied to the text and it has been greatly improved, and I thank the respected authors for their great care and attention to detail. However, some parts of manuscript still need to be revised, the most important of which are shown in the attached PDF file. It is better that the requested corrections are taken into consideration before accepting this article for publication.

Author Response

esponse:Thank you for your further suggestions on the manuscript. I have modified it according to your suggestions and PDF,including the figure of pathogenicity, morphological characteristics  and  phylogeny, some language description and details. Hopefully, this study is updated and strong enough to be accepted by the Journal of Fungi.